# Risk Assessment of the Dietary Phosphate Exposure in Taiwan Population Using a Total Diet Study

**DOI:** 10.3390/foods9111574

**Published:** 2020-10-30

**Authors:** Min-Pei Ling, Jun-Da Huang, Huai-An Hsiao, Yu-Wei Chang, Yi-Ting Kao

**Affiliations:** 1Department of Food Science, National Taiwan Ocean University, Keelung City 20224, Taiwan; huangjunda0807@gmail.com (J.-D.H.); schwarm331@gmail.com (H.-A.H.); bweichang@email.ntou.edu.tw (Y.-W.C.); 2Food and Drug Administration, Ministry of Health and Welfare, Taipei City 115209, Taiwan; fskaoit@gmail.com

**Keywords:** calcium, intake, phosphate, phosphorus, risk assessment, total diet study

## Abstract

Phosphorus and calcium are essential nutrients for the human body. However, excessive intake of phosphates and a low calcium:phosphorus ratio can lead to disorders in calcium-phosphorus metabolism, kidney disease, or osteoporosis. In this study, a total diet study (TDS) was used. The total phosphorus concentrations of foods were combined with the average dietary consumption to calculate the estimated daily intake, which was compared with the maximum tolerable daily intake (MTDI) to assess the resulting health risk of total phosphorus exposure. The calcium concentration in food and total calcium intake were also analyzed and estimated to calculate the calcium:phosphorus ratio. In conclusion, the phosphate exposure risks for the Taiwanese population are acceptable. However, the calcium:phosphorus ratio in the Taiwanese population (0.51–0.63) is lower than the reference calcium:phosphorus ratio (1.25).

## 1. Introduction

Phosphorus is an essential nutrient and component of the human body. It is used in the form of phosphate and plays an important role in the metabolism of carbohydrates, fats, and proteins. Most phosphorus intake comes from food. There are two main types of phosphorus in dietary products: natural and added phosphorus. Because of the difference in chemical properties and physiological characteristics, both the absorption rate and absorption efficiency are different. Added phosphorus is mainly inorganic phosphorus added to food products during the preparation process to keep freshness, stabilize quality, and extend preservation; it does not need to be digested or degraded by enzymes and can be quickly dissociated in the acidic environment of the stomach. The absorption efficiency and bioavailability are extremely high and so the absorption rate of added phosphorus can reach up to 80–100%. In contrast, the absorption of natural phosphorus is slow and the bioavailability is lower, with an absorption rate ranging from only 10–60% [1]. One study demonstrated that a high-phosphorus–low-calcium diet is followed in Taiwan [2]. Healthy people and animals have a higher tolerance for phosphate, but excess phosphorus intake is harmful to the human body and may lead to elevated serum phosphorus, triggering calcium–phosphorus metabolism disorders, osteoporosis, chronic kidney disease, and cardiovascular diseases [3,4,5]. Phosphate additives play important roles as acid regulators, emulsifiers, humectants, leavening agents, sequestrants, stabilizers, and thickeners in the food industry. Phosphate is commonly used in monophosphate, polyphosphate, and pyrophosphate forms. Studies in recent years have discovered that more attention has been paid to phosphorus intake along with the widespread use of phosphate additives, which is likely to further cause calcium: phosphorus ratio imbalances. One Belgian study compared the phosphorus content of local poultry products with the values for the preserved poultry products found in the European Union database. They found that the phosphorus content per 100 g of chicken was 84 mg higher than that reported in the database [6]. The phosphorus content of processed meat products was 28% higher than freshly slaughtered meat [7]. Meat products may have added phosphate [8,9]. In the United States, >50,000 foods contain one or more added phosphates. Food used in restaurants and fast-food chains often have added phosphates, causing people to consume more phosphorus. Therefore, the actual intake of phosphorus may be higher than the phosphorus content in the food composition database [10,11]. The total phosphorous content of convenience food with added phosphate is 70% higher than that of products without added phosphate [12]. Risks from excessive phosphorus consumption are likely to exist. Excessive phosphorus dietary intake is likely to increase serum phosphorus, cause imbalances in the calcium:phosphorus ratio, and elevate the risk for cardiovascular disease [13]. The surge in parathyroid hormone (PTH) concentration induced by a high-phosphorus diet is considered to have positive correlations with diseases or syndromes of osteoporosis and low bone density triggered by ageing [14]. Therefore, conducting a risk assessment for estimating the actual intake of dietary phosphorus is necessary.

This study mainly explores the health risks related to the actual intakes of phosphate in the Taiwanese population, using a total diet study (TDS) to assess the estimated daily intake (EDI) of phosphates. The hazard quotient (HQ) was assessed based on the consumption of phosphates by the Taiwanese population. After the percentage of EDI over the maximum tolerable daily intake (MTDI) (%MTDI) was calculated, an analysis was conducted to explore the risk of phosphate intake in each exposure group. A high-phosphorus–low-calcium diet is considered to have a positive correlation with the occurrence of osteoporosis. Therefore, the consumption data for the elderly group in the Nutrition and Health Survey in Taiwan (NAHSIT) database [15,16,17] was combined with the food product sample concentration data of the TDS, and the EDI and dietary calcium:phosphorus ratio for this group was calculated to assess risk.

## 2. Materials and Methods

### 2.1. Classifying and Clustering Food Products

The study relied on the data of a 1-day 24-h drinking and eating review in the NAHSIT from 2005 to 2008, 2010, 2011, and 2012. There were a total of 8025 respondents [15,16,17]. After excluding the responses where physical examination data did not provide weight information and data for infants (0–1 years), who are less likely to consume foods with added phosphate, 7580 respondents met the requirements for this study. According to the “The Preliminary draft Amendment of Standards for Scope, Application and Limitation of Food Additives, Second Edition,” which was formulated by the Taiwan Food and Drug Administration (TFDA), the NAHSIT raw data was clustered into 16 main categories and 285 food subcategories. Subsequently, the consumption amount of food and the proportion of people consuming each food product in each exposure group was calculated to create a representative food product list for phosphates. The exposure groups were divided according to the recommendations of EFSA in the risk assessment of food additives [18] into: toddlers (1–2 years), children (3–9 years), adolescents (10–17 years), adults (18–64 years), and the elderly (over 65 years old); except for the toddler group, each age level was further subdivided into males and females, along with the whole group taken together. The total number of exposure groups came to 13.

### 2.2. Representative Food Product List

Figure 1 shows the flowchart of implementation. Depending on the raw material sources used in the food and if the processing and manufacturing methods were similar or not, the purchased food list was divided into a national food list and a regional food list. In the representative food product list, if the brand name proportion of a food item accounted for more than 50% or less than 50% but belonging to raw materials (e.g., sugar, salt, and ice cubes), then it was categorized as a national food product. To obtain a national representative food product list, sorting was divided into the following scenarios:Scenario 1: If the total proportion of food consumption was ≥50% and if these items had well-known brand names, then they were listed as purchased items.Scenario 2: If the total proportion of food consumption was ≥50%, but they were from unknown brand names, then the food products were ignored. Eight food products with brand names were selected and sorted out in descending order based on the food consumption proportions.Scenario 3: If the items were all unknown brand names, then eight sets of food products were randomly purchased in the market.

If the brand name proportion of a food item was less than 50% and they did not belong to raw materials, then it was categorized as a regional food product. To obtain a regional representative food product list, the sorting was divided into the following scenarios:Scenario 1: If the total proportion of food consumption amount for a whole group was ≥50% and it had multiple items of different types, then those having similar total phosphorus content were listed as purchased items.Scenario 2: If the total proportion of food consumption amount for a whole group was ≥50% and it had multiple items with different or unknown total phosphate, then they were listed as individually purchased.

### 2.3. Sampling and Preparing Samples

The national food products were nationwide samples depending on the market brand name proportions. Food items were purchased at supermarkets and convenience stores in Keelung City. The regional food products were samples taken from the north, west, south, and east regions in Taiwan. The most populated county or city and a random county or city among each region were selected. A total of eight counties or cities (New Taipei City and Keelung City in the Northern region; Taichung City and Changhua County in the Central region; Kaohsiung City and Tainan City in the Southern region; and Hualien County and Taitung County in the Eastern region) were sampled in traditional markets, restaurants, and night markets between September and December 2016. After the sample collections were completed, 876 purchased food products were classified according to the 285 food subcategories. After cooked or processed to ready-to-eat, food products were separately mixed and homogenized into 168 samples for analysis (Figure 1).

### 2.4. Analyzing the Total Phosphorus and Calcium Concentration in Samples

The samples were processed by microwave-assisted acid digestion and injected into an inductively coupled plasma optical emission spectrometer (ICP-OES; Agilent 5100, Santa Clara, CA, USA) to analyze the content of phosphorus and calcium based on the “General Method of Test for Heavy Metals” of Taiwan [19]. When the sample concentration was undetected, meaning that it was below the limit of detection (LOD), and if the number of undetected samples was less than or equal to 60%, then LOD/2 was substituted for samples with a concentration below the LOD. If the number of undetected samples were more than 60% but less than or equal to 80% and when the number of detected samples was at least 25, then the upper bound (UB) of the undetected sample was expressed as the numerical value of the LOD and the lower bound (LB) was expressed as 0. If the number of undetected samples was more than 80%, then the UB of undetected samples was represented by the numerical value of LOD and the LB was represented by 0.

### 2.5. Total Phosphorus and Calcium Intake and Risk Assessment

Using the NAHSIT data, the representative food product, food consumption, and weight of each exposure group was assessed. In accordance with the aforementioned sampling method, the concentration of phosphorus and calcium was measured after sampling, cooking, and homogenizing the product. Food consumption was calculated to assess the intake of total phosphorus and calcium in each exposure group using the following equation:(1)$${\mathrm{EDI}}_{\mathrm{ij}}=\sum_{i=1}^{n}\frac{C_{\mathrm{ij}}\times {\mathrm{CR}}_{\mathrm{ij}}}{{\mathrm{BW}}_{j}}.$$

In Equation (1), EDI_ij_ is the EDI of phosphate or calcium in food i consumed by exposure group j (mg/kg bw/day), C_ij_ is the phosphate or calcium concentration in food i (mg/kg), CR_ij_ is the consumption rate of food i for each exposure group j (kg/day), and BW_i_ is the body weight of each exposure group j (kg).

Phosphorus (mainly as phosphates) is both a necessary nutrient for the human body and a common ingredient in food. Hence, a Joint FAO/WHO Expert Committee on Food Additives (JECFA) considers that for measurements of the toxicity of phosphates, it is not appropriate to set up an acceptable daily intake (ADI); instead, the MTDI should be taken as a measurement standard. The JECFA conducted a survey and collection of toxicological data on commonly seen phosphates in the 1970s, taking the lowest dose for renal calcium deposits in rats (1% phosphorus (P)) as the evaluation basis and based the daily food intake of 2800 calories as an estimation. The MTDI value of P is calculated at 70 mg/kg bw/day [20]. The amount of food additives used is calculated by measuring the P content (70 mg P equivalent to the total P of 160 mg of P_2_O_5_). In this study, the EDI and MTDI values of foods to be assessed were compared to assess the hazard index of phosphates in the diet as expressed in %MTDI. The %MTDI is the exposure to food additives for each exposure group through food consumption and can be calculated through the following equation:(2)$${\%\mathrm{MTDI}}_{\mathrm{ij}}=\frac{{\mathrm{EDI}}_{\mathrm{ij}}}{\mathrm{MTDI}}\times 100\%$$

In Equation (2), %MTDI_ij_ is the hazard index, EDI_ij_ is the EDI of phosphates by exposure group j from food i (mg/kg bw/day), and MTDI is the maximum tolerable daily intake of phosphates (70 mg P/kg bw/day).

The representative food product and food consumptions of each exposure group were assessed, according to the NAHSIT database. Calcium and phosphorus intakes for each exposure group were assessed and compared with the adequate intake (AI) and tolerable upper intake level (UL) available in the seventh edition of the “Dietary Reference Intakes of Taiwanese” published by the Health Promotion Administration, Ministry Health and Welfare in Taiwan to assess the risk of the population intake of calcium and phosphorus. [21] The AIs of phosphorus is different for each group: 1–3 years (400 mg P), 4–6 years (500 mg P), 7–9 years (600 mg P), 10–12 years (800 mg P), 13–18 years (1000 mg P), and 19+ years (800 mg P). The ULs of phosphorus are different for groups: 1–9 years (3000 mg P), 10–70 years (4000 mg P), and 71+ years (3000 mg P). The AIs of calcium are different for groups: 1–3 years (500 mg Ca), 4–6 years (600 mg Ca), 7–9 years (800 mg Ca), 10–12 years (1000 mg Ca), 13–18 years (1200 mg Ca), and 19+ years (1000 mg Ca). The UL of calcium for all groups is 2500 mg Ca.

## 3. Results and Discussion

### 3.1. Calcium and Phosphorus Intake and Potential Health Risks

Intake scenario of phosphorus: as shown in Figure 2, the phosphorus intake of the population ranged from 1122.5 to 1511.8 mg P/day. The group with the highest phosphorus intake was the toddler group, with an intake from 1452.0 to 1511.8 mg P/day. The group with the lowest phosphorus intake was the group of children, with an intake from 1122.5 to 1345.6 mg P/day. The phosphorus intakes of all groups were higher than the AI but lower than the UL value, so there was no significant risk. The solid bars represent the minimum intake, and the hollow bars represent the maximum intake in Figure 2. The intake of phosphorus shows an increasing trend compared with the past. As reported in one study, the daily phosphorus intake of adults aged 19–64 years old between 1993 and 1996 was 1087 mg for males and 858 mg for females [2]. Wu et al. [2] and this study show that the main sources of dietary phosphorus for adults aged 19–64 years old were broken and flaked grains and fresh raw meats from livestock, poultry, and exotic animals, indicating that the dietary structure of the population has not changed fundamentally in recent years. The reason phosphorus intake has increased is possibly related to changes in the dietary proportion. As indicated in the dietary and nutritional status of Taiwan announced by the Ministry of Health and Welfare, the protein intake of the population at the current moment has grown significantly compared with the period between 1993 and 1996. This may be the reason for the increase in the phosphorus intake of the population. Moreover, the widespread use of phosphates cannot be ruled out as a cause of the increase in total phosphorus content in the diet.

Intake scenario of calcium: as shown in Figure 2, the calcium intake of the population ranged from 493.6 to 2199.8 mg Ca/day. The group with the highest calcium intake was the toddler group, with an intake of 1565.8–2119.8 mg Ca/day. The group with the lowest calcium intake was the adolescent group, with an intake of 493.6–680.3 mg Ca/day. The solid bars represent the minimum intake, and the hollow bars represent the maximum intake in Figure 2.

Even though the calcium intake has increased when compared with a previous study, it is still unable to meet the AI. Insufficient calcium intake is a common problem around the globe. In Shenzhen, China, calcium intake in 2010 was 620 mg Ca/day [22]. In France, calcium intake by TDS in 2005 was 721 mg Ca/day [23]. In Cameroon, the calcium intake in 2013 was 760 mg Ca/day [24]. Calcium intake in all these countries or regions have not reached the recommended intake of their local setting. High-phosphorus–low-calcium diets may lead to an elevated risk of reduced bone density. Diets with a low calcium:phosphorus ratio may have negative effects on bones because a high phosphorus intake may lead to a long-term rise in PTH concentration, leading to a decrease in osteocalcin content and bone density.

### 3.2. Main Contributing Food Products of Phosphate Exposure

As shown in Figure 3, the highest average concentrations of phosphorus contributing to the intake came from products intended for particular nutritional uses (5016.1 mg/kg), followed by processed snacks and nut products (4950.3 mg/kg), fish and fish products (3268.9 mg/kg), as well as dairy products and their analogues (3187.5 mg/kg). The lowest concentration was found in sweeteners, including honey (33.7 mg/kg). The concentration of phosphorus and calcium in each food subcategory can be found in Appendix A.

In this study, the food subcategories in which the individual EDI consumed by each exposure group was greater than or equal to 5% was listed as the main contributing food item. As shown in Figure 4, the main dietary sources of phosphorus in all age groups (e.g., whole, broken, or flaked grains, including rice; fresh meat and poultry; as well as milk powder and cream powder) were foods without added phosphates, mainly containing natural phosphorus. Thus, it can be estimated that natural phosphorus is the main source of phosphorus intake in the diet of the Taiwanese population.

### 3.3. Safety Risk Assessment of Toddler Food Products

The JECFA used the lowest dose for renal calcium deposit in rats (1% P) as the endpoint; the human body parameter was reported to beat 70 mg/kg bw/day [20]. Taking 70 mg/kg bw/day as a safety reference value, the phosphorus intake of each age group is shown in Table 1. As described in the table, the EDI of total phosphorus intake ranged from 18.4 to 122.9 mg P/kg/day. The female adult group had the lowest intake, from 18.4 to 21.5 mg P/kg/day. The toddler group had the highest intake, from 118.1 to 122.9 mg P/kg/day. The hazard index of phosphates assessed in the diet was expressed as %MTDI. The %MTDI of all whole groups ranged between 26.3% and 175.6%. The %MTDI of the toddler group ranged between 168.9% and 175.6%, which was the highest among all whole groups; the %MTDI of the adult group ranged between 26.3% and 35.3%, which was the lowest among all whole groups. There was no significant risk for other groups, except for the toddler group. Although the absorption rate of added phosphorus can reach 80–100%, the absorption of natural phosphorus is slow and the bioavailability is lower, at only 10–60% [1]. Natural phosphorus accounts for most of the phosphorus in food products for toddlers, which makes up 76% of this study. It can be inferred that the actual intake of phosphorus in toddlers is much lower than the theoretical calculation value, the result is approximately 105–109%MTDI.

According to a relevant study by the Institute of Medicine (US), the serum phosphorus concentration of infants and toddlers fed with formula was not different from breast-fed infants and toddlers, and no abnormalities in body function has been found [25]. As considered by this study, the phosphorus excretion mechanisms of infants’ and toddlers’ kidneys can withstand a wide range of phosphorus intake to support growth and development.

Using %MTDI to assess risks is a highly conservative assessment. The conservative principle is an intrinsic feature in this type of safety assessment. This can ensure that human health receives adequate protection. The MTDI for phosphorus was laid down by the recipe from a daily dietary intake of 2800 kcal and phosphorus intake not exceeding 6600 mg. The daily calorie intake of the Taiwanese population rarely reaches 2800 kcal. According to the “Nutritional Reference Manual for Toddlers (draft),” the daily caloric intake of toddlers was only 1355 kcal [26]. In terms of phosphorus intake, toddlers’ intake calculated in this study ranged from 1452.0 to 1511.8 mg, which was far from the phosphorus intake of 6600 mg. Moreover, the risk defined by MDTI is a risk of developing adverse effects after continuously ingesting nutrients at this concentration for over 70 years; in fact, it is difficult to achieve the conditions for maintaining continuous intake at this concentration for 70 years. Therefore, it is reasonable to believe that phosphorus intake for the toddler group aged 1 to 2 years is safe. The risk in this group is within an acceptable range.

### 3.4. Survey of Calcium and Phosphorus Intake in the Elderly Population

The bone density of the elderly decreases with age, and thus, they become more susceptible to developing osteoporosis. Animal studies have shown that feeding experimental animals a high-phosphorus diet with an unbalanced calcium:phosphorus ratio can induce secondary hyperparathyroidism, bone resorption, low peak bone mass, and fragile bones [27]. Another study indicated that an increase in PTH concentration induced by a high-phosphorus–low-calcium dietary intake has a positive correlation with osteoporosis and low bone density due to aging [12]. Therefore, the following discussion analyzes the calcium:phosphorus ratios in the elderly at high risk for osteoporosis in order to quantify the related risks.

As shown in Figure 5, (A) 69.3% of adult males over the age of 60 years old had calcium intakes below 1000 mg, 68.4% of adult males from 50 to 60 years old had a calcium intake below 1000 mg, and adult females from the same age range had higher calcium intakes by 11.5% and 11.3%, respectively. (B) Phosphorus intakes are lower than 800 mg in 11.1% of adult males and 25.8% of adult females over the age of 60 years, in 8.4% of adult males, and in 22.6% adult females between 50 and 60 years of age. (C) The calcium:phosphorus ratios calculated and obtained from the AI of phosphorus and calcium showed that 99.1% of males over 60 years, 98.4% of females over 60 years, 99.3% of male adults 50–60 years, and 97.7% of females of 50–60 years had ratios lower than the reference calcium:phosphorus ratio of 5:4.

In the past few decades, the intake of phosphorus has increased more drastically than the intake of calcium, which has also led to the status quo of a current imbalance in the calcium:phosphorus ratio. The calcium:phosphorus ratio among the population is from 0.38 to 1.40. The group with the highest calcium:phosphorus ratio was the toddler group with a ratio of 1.12–1.40; the group with the lowest ratio was the male adolescent group, with a ratio of 0.38–0.45. This paper assumed that the recommended calcium:phosphorus ratio is calculated from the AI value. Compared with the recommended calcium:phosphorus ratio, the ratios of the toddler group are close to the recommended value, but larger gaps between the calcium:phosphorus ratio and the recommended ratio were found for the other groups. As reported by Kemi et al., cross-sectional studies of young Finnish females showed that the calcium intakes of one-fourth of young females had not reached the recommended daily intake (RDA), and their phosphorus intakes are twofold the RDA value, thus the calcium:phosphorus ratio was 0.56 [12]. The concentration of serum PTH in this group was significantly higher than in the other three-fourth of the women (calcium: phosphorus ratio greater than 0.7). Long-term high-phosphorus–low-calcium diets are considered to have a direct correlation with an elevated risk factor for bone fractures and osteoporosis.

Data from 2002 revealed that the proportion of osteoporosis in Taipei female patients rises sharply after they reach menopause (at about 50 years old). When males are between 60 and 65 years old, a similar phenomenon occurs [28]. Even though the calcium:phosphorus ratio of elderly females is higher than that of males, the incidence of osteoporosis is higher in females, according to the analysis of epidemiological results. According to the 2005 statistics of the Bureau of Health Promotion of Taiwan, about 16% of the population over the age of 60 suffered from osteoporosis, 80% of which were females. In addition, as pointed out by the study results [29] and according to the results of the 2005–2006 National Health and Nutrition Examination Survey (NHANES), the calcium:phosphorus ratio of elderly females is higher than that of elderly males and some females meet the recommended calcium:phosphorus ratio. The probability that elderly females may develop osteoporosis symptoms is still significantly higher than that of elderly males, and both the fracture rate and mortality rate of elderly males is lower than for elderly females. Based on this finding, the risk assessment for bone fractures in the elderly using the calcium:phosphorus ratio requires further studies. However, a previous study pointed out that phosphorus intake higher than 1400 mg is considered to correlate with higher all-cause mortality [30].

### 3.5. Uncertainty Analysis

Situational uncertainty: There is uncertainty in the food product matching and clustering, especially in compound food products, and there are possibilities for dose overestimation or underestimation. Estimating the long-term food consumption using short-term dietary data may overestimate food consumption. The food collection in this study is based on a representative food product list of phosphates, so the analyzed calcium concentration and calcium intake may not fully reflect the actual situation in Taiwanese population, which is a limitation of this study.

Parametric uncertainties: The food consumption data of the NAHSIT originates from questionnaire surveys. Respondents may have bias in their own judgements on food consumptions, resulting in uncertainty in the calculation of exposure dose. The food subcategories with more detailed characteristics in the “The Preliminary draft Amendment of Standards for Scope, Application and Limitation of Food Additives, Second Edition” cannot be obtained from the raw data of the NAHSIT. If the concentration value used has restrictions in terms of details, it can cause an overestimation of exposure dose.

Model uncertainty: In this study, the absorption rate of phosphorus is calculated as 100% absorption, and there is a difference in the absorption of organic phosphorus and essential inorganic phosphorus, which leads to an overestimation of risk. MTDI is a recommended daily intake per person for life. Taking phosphorus as an example, even if the daily intake of phosphorus per person is less than or equal to 70 mg/kg bw/day, there will be no risk. However, lifetime consumption at this dosage is merely possible theoretically, so the risk may be an overestimation. In the 1-day, 24-h dietary review method adopted by the NAHSIT database, individual bias cannot be corrected, this may cause overestimation of the risk of the high-exposure group.

## 4. Conclusions

Using the TDS method, this study explored the risk of phosphate intake in the Taiwanese population. A total of 168 food product samples were collected. By combining the phosphorus concentration of samples with the food consumption data from the NAHSIT database, the phosphate intake of population and their risks were calculated. The phosphorus intakes in each whole group were 46.8–56.0 mg P/kg/day–toddlers (1–2 years), 46.8–56.0 mg P/kg/day—children (3–9 years), 22.1–26.2 mg P/kg/day—adolescents (10–17 years), 19.9–23.1 mg P/kg/day—adults (18–64 years), and 21.6–24.3 mg P/kg/day—the elderly (65+ years). Under the scenario of highly conservative estimates and considering factors such as absorption rate and metabolism scenarios, the risks from phosphate intake by group populations are within an acceptable range. Phosphorus intake of the population is adequate, while calcium intake has not reached sufficient intake. For the intake of calcium, only the toddler group (1–2 years) and children (3–9 years) meet the AI. The population should pay close attention to calcium supplements and reduce consumption of high-phosphorus foods in order to maintain the balance between calcium and phosphorus. The risk of a high-phosphorus–low-calcium diet for the elderly cannot be ignored. Therefore, the elderly should actively try to prevent the occurrence of osteoporosis, pay attention to their dietary calcium:phosphorus ratio, and appropriately supplement with calcium and vitamin D.

## Figures and Tables

**Figure 1 foods-09-01574-f001:**
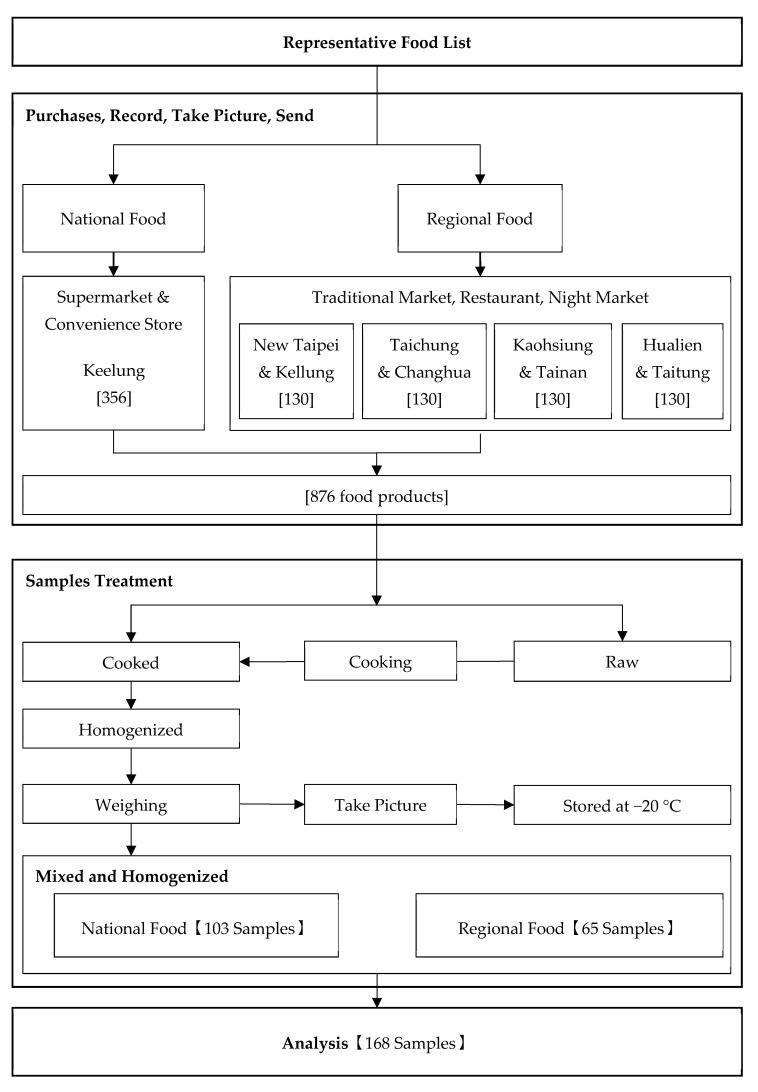
Sample implementation flowchart.

**Figure 2 foods-09-01574-f002:**
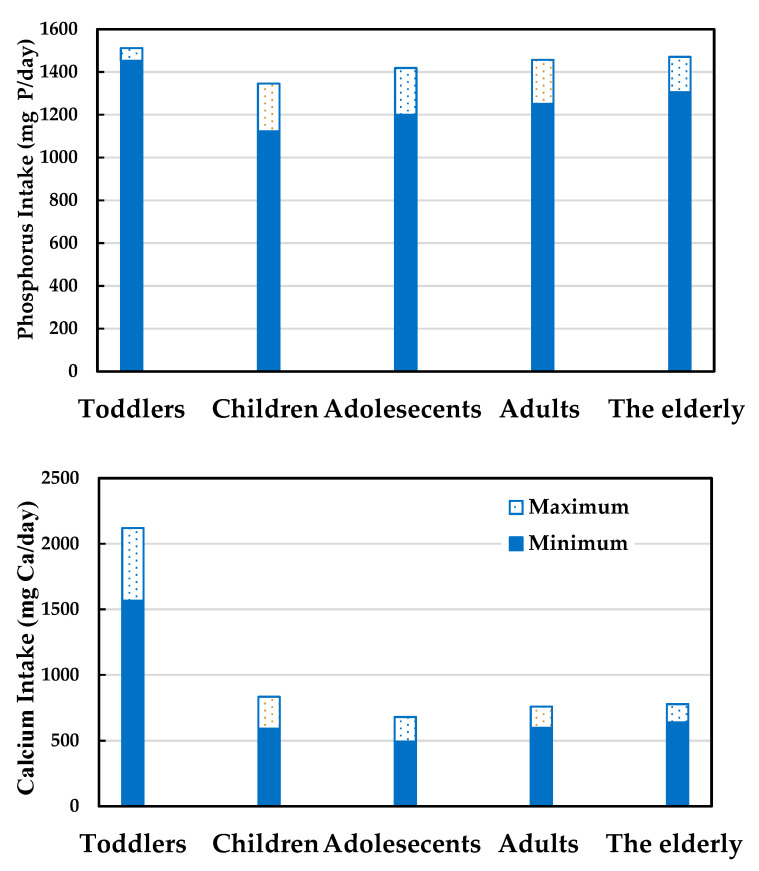
Comparison of calcium and phosphorus intake between exposure groups.

**Figure 3 foods-09-01574-f003:**
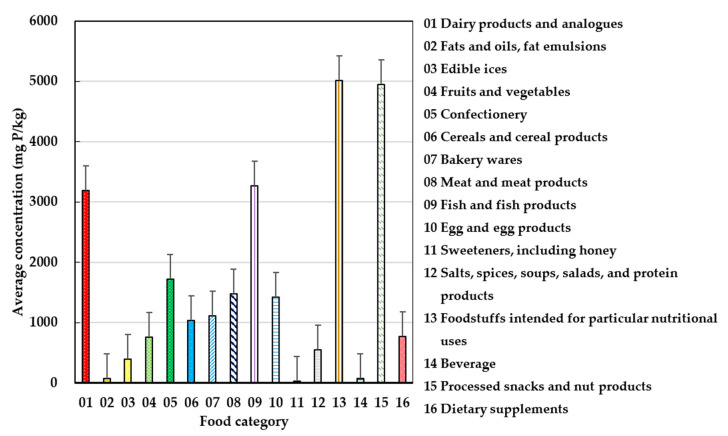
Comparison of the phosphorus concentrations for 16 food categories.

**Figure 4 foods-09-01574-f004:**
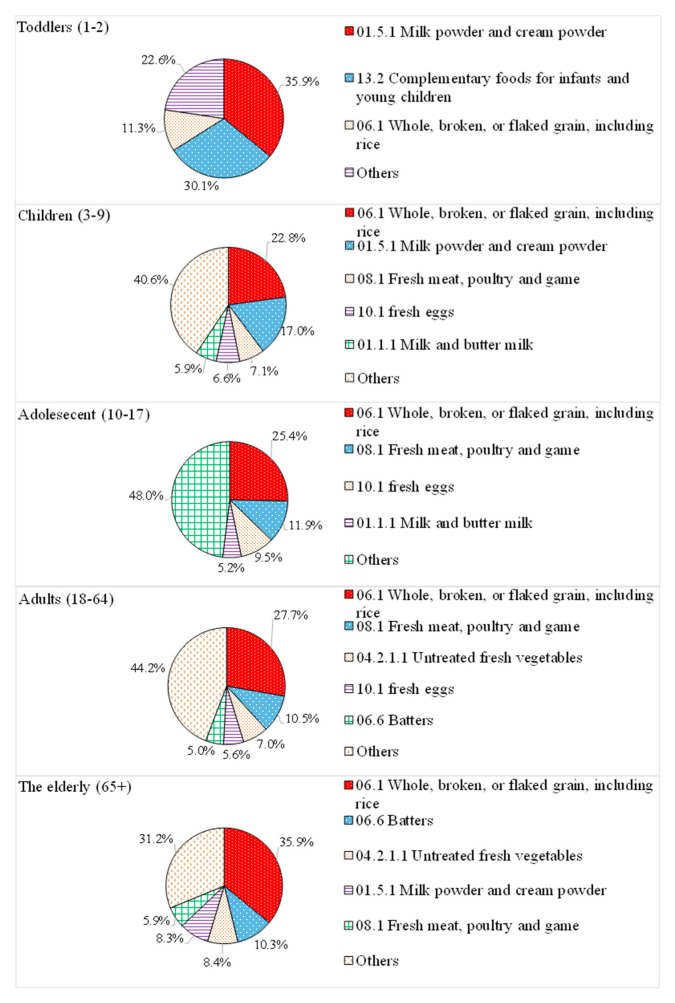
Contribution of food groups to dietary intake of phosphorus in whole groups, expressed in percentages. The numbers in front of food groups are the number of food subcategories.

**Figure 5 foods-09-01574-f005:**
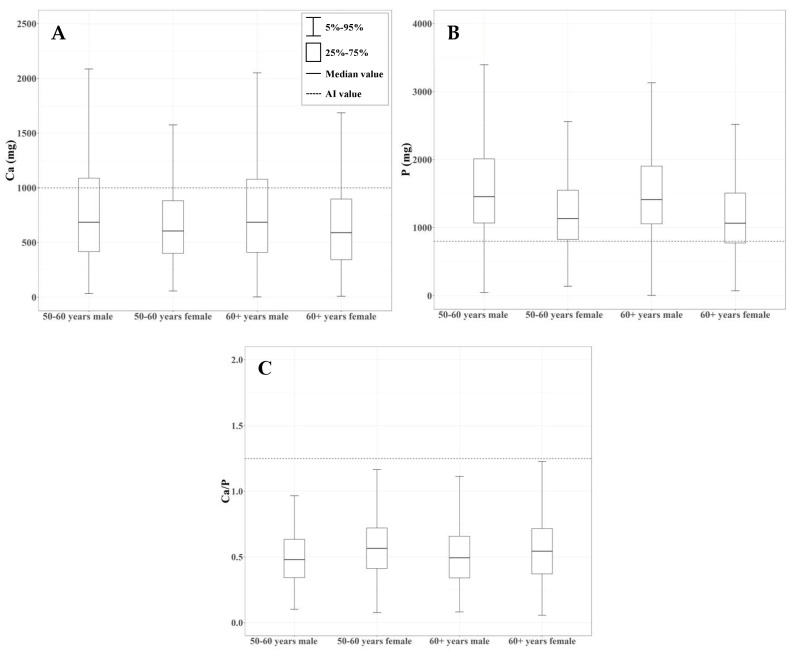
Summary and distributions of (**A**) total calcium intake with recommended daily intakes (RDAs). (**B**) Total phosphorus intake with RDAs. (**C**) Estimated Ca:P ratio. Box plots show the distribution of calcium, phosphorus, and Ca:P ratio by gender and age group.

**Table 1 foods-09-01574-t001:** Hazard index (percentage of estimated daily intake over the maximum tolerable daily intake, %MTDI) for 13 age/gender subgroups in Taiwan.

Exposure Groups (Years)		EDI (mg P/kg/day)	%MTDI
**Toddlers** **(1–2)**	**Whole**	118.1–122.9	168.7–175.6
**Children** **(3–9)**	**Male**	49.4–59.0	70.54–84.3
	**Female**	44.2–53.2	63.2–76.0
	**Whole**	46.8–56.0	66.8–80.0
**Adolescents** **(10–17)**	**Male**	24.0–28.3	34.3–40.4
	**Female**	20.5–24.4	29.3–34.9
	**Whole**	22.1–26.2	31.6–37.4
**Adults** **(18–64)**	**Male**	21.3–24.7	30.4–35.3
	**Female**	18.4–21.5	26.3–30.7
	**Whole**	19.9–23.1	28.4–33.0
**The elderly** **(65+)**	**Male**	22.7–25.8	32.4–36.9
	**Female**	20.4–22.7	29.1–32.4
	**Whole**	21.6–24.3	30.8–34.7

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
