# Peer review of "Risk Assessment of the Dietary Phosphate Exposure in Taiwan Population Using a Total Diet Study"

_foods, 2020, doi:10.3390/foods9111574_

Round 1

Reviewer 1 Report

Thank you for your comments.

Author Response

Thank you very much.

Reviewer 2 Report

The authors have addressed almost all my comments clearly.

However, there are still a a couple of unanswered comments that should be corrected.

In paragraph 2.1, I would like the 16 food categories to be stated (Comment 1).

The figure legend for Figure 5 has not been revised (only RDA introduced) and the figure does not depict histograms but box plots. This should be corrected. 

When the last paragraph of the introduction was shortened the abbreviation NAHSIT was mistakenly not introduced in this version of the manuscript, this should be corrected.

Author Response

We appreciate the reviewer for providing these valuable comments. The comments could greatly improve the quality of this manuscript. The manuscript has been revised with red markings to highlight the changes and new entries. The followings are the detailed responses to each individual comment.

1. In paragraph 2.1, I would like the 16 food categories to be stated (Comment 1).

Response: The 16 food categories in this study were referenced from the food categories of“Preliminary Draft of Standards for Use Scope and Limitations of Food Additives in Taiwan, Second Edition”, which was the draft regulations of food additives in Taiwan. We have slightly modified the text (P2, L76-78). The detailed of 16 food categories is shown in the legend in Figure 3 (P7, L228) and (Appendix).

2. The figure legend for Figure 5 has not been revised (only RDA introduced) and the figure does not depict histograms but box plots. This should be corrected.

Response: Thank you for correction. Figure 5 has been revised histograms into box plots (P10, L296-297).

3. When the last paragraph of the introduction was shortened the abbreviation NAHSIT was mistakenly not introduced in this version of the manuscript, this should be corrected.

Response: Thank you for correction. The abbreviation of Nutrition and Health Survey in Taiwan (NAHSIT) was corrected in the manuscript (P2, L66-67).

Reviewer 3 Report

Before the proper review please provide the supplementary material with the data on the phosphorus content determined in 168 samples using ICP-OES.

Author Response

We appreciate the reviewer for providing these valuable comments. The comments could greatly improve the quality of this manuscript. The manuscript has been revised with red markings to highlight the changes and new entries. The followings are the detailed responses to each individual comment.

General comments

1. The manuscript should be carefully read and corrected. Many grammar and style corrections are required, e.g. lines 10, 16, 18, 61, 64, 139, 218 etc. Please use the past tense throughout the M&M and R&D sections.

Response: Thank you for correction. We have checked the grammar and corrected it.

2. Lines 127-130: how did you combine 876 into 168 samples? Does it mean that in Supplementary Material the g line “ice-cream” was calculated from several products or was measured for one sample (which was a mix of different products)?

Response: In this study, several food products of the same type subcategory were mixed multiple into 1 sample, e.g., 8 different brands of ice cream mixed into 1 sample: 03.1 Ice cream; another 8 different brands of popsicle mixed into 1 sample: 03.2 Popsicle.

Detailed comments

3. Line 14: change ‘maximum tolerable daily intake’ into ‘provisional maximum tolerable daily intake’ or ‘tolerable daily intake’. Please change also the appropriate abbreviations (e.g. line 61).

Response: The abbreviations of MTDI in abstract is revised (P1, L14). In JECFA (1970) literature, maximum tolerable daily intake (MTDI) of 70 mg/kg body weight (bw) for phosphate salts , expressed as phosphorus are used. We decided to refer JECFA and use MTDI

4. Line 17: change the sentence into “the is no risk of the excess of phosphorus in the (…).

Response: Because of the uncertainty of research, we cannot be completely certain that there is no risk, so we conclude the risks of phosphate are acceptable.

5. Line 18: in bracket add the ratio for Taiwanese population.

Response: We added the range of Ca/P ratio for Taiwanese population in (P1, L17-19).

6. Line 20: please reformulate key words into e.g.: calcium, intake, phosphorus, risk assessment, Taiwan, total diet study

Response: Thank you for kind comment. We have reformulated key words into phosphate; phosphorus; calcium; intake; risk assessment; total diet study (P1, L20).

7. Line 29: what is the difference between ‘processing’ and ‘preparation process’? Please reformulate this sentence.

Lines 30 and 31: change semicolons into stops.

Response: Thank you for kind comment. This paragraph was revised to “Added phosphorus is mainly inorganic phosphorus added to the food products in the processes and preparation to keep fresh…”(P1, L28-29). Semicolons was changed into stop (P1, L31).

8. Line 36: what does it mean ‘normal’?

Response: The word normal people was changed into Healthy people (P1, L35-36).

9. Lines 44-49 and references [6,7]: please add/replace with more up-to-date studies on the phosphorus excess (e.g. of Borgi, 2019; https://doi.org/10.2215/CJN.07230618).

Response: Added up-to-date studies of phosphate “Meat products may be added with phosphate, causing people to consume more phosphorus, the actual intake of phosphorus may be higher than the phosphorus content in the food composition database” in (P2, L49-51; P13, L413-418).

10. Line 64: add the reference.

Line 64 and paragraph 2.1: please add the information that NAHSIT was conducted among Taiwanese population. If it is an abbreviation please expand it.

Response: Thank you for correction. The reference has been added (P2, L66-67). The abbreviation of Nutrition and Health Survey in Taiwan (NAHSIT) was corrected in the manuscript (P2, L66-67).

11. Line 91: add names of the regions from Figure 1 to the text.

Response: Names of cities of 4 regions from Figure 1 listed in (P3, L126-129).

12. Lines 116-127 should be combined with lines 84-93. They should be also shorten.

Response: Paragraph 2.2 established representative food product list, than paragraph 2.3 Sampling and preparing samples, there were 2 different methods.

13. Figure 1: cross out (1), (2), (3) in the description of markets and restaurants.

Response: The description of figure 1 has been modified (P4, Figure 1).

14. M&M: as the [15] is in Taiwanese language, please add the short description for mineralization process and ICP-OES analysis.

Response: The samples were processed by microwave-assisted acid digestion and injected into inductively coupled plasma optical emission spectrometer (ICP-OES) (Agilent 5100, California, US) to analyze the content of phosphorus and calcium with the method based on “General Method of Test for Heavy Metals” of Taiwan (P4, L138-141).

15. The supplementary material: 1/ please add SD values or ranges, 2/ add data on the calcium content, 3/ prepare it with an adequate scrupulosity.

Response: We list the concentration of phosphorus and calcium in (Appendix), but no SD data due to research limitations (only 1 concentration for 1 food subcategory). This study using “total diet study” method, the purpose is to calculate the dietary exposure of the population and its potential risks to public health on a rigorous basis. To assess all the foods people in Taiwan might be intake, we have to ensure the food groups was most people eaten. Based on it, the concentration of each food subcategory which mixed from 8 regions of Taiwan could represent the average value of all nationals.

16. Line 149: what does it mean “measured again”?

Response: Thank you for kind comment. We have revised the text “In accordance with the aforementioned sampling method, the concentration of phosphates and calcium were measured after sampling, cooking, and homogenizing.” to make clearly (P5, L151-152).

17. Figure 2: where does the range of intakes come from? From different years presented in NIHST?

Response: In this study, foods with larger differences in texture will be analyzed separately, e.g.,

01.5.1 Milk powder and cream powder-Skimmed milk powder as 1sample, 01.5.1 Milk powder and cream powder-Growth milk powder as 1 sample, 01.5.1 Milk powder and cream powder-Milk powder as 1 sample. Calculating the risk with different concentrations will output different range of risk. Concentration of 168 samples can be seen in (Appendix).

18. Figure 4: please cross out the numbers neat food groups (01.5.1, 13.2, (…)

Response: For the convenience of corresponding food categories, we recommend keeping the category numbers.

19. Line 348: cross out the ‘the risk’.

Conclusions: please significantly shorten the conclusions (by omitting methodological data in lines 348-353; by shortening long sentences; by omitting general statements). Moreover, please add the concrete numbers for phosphorus intakes in different ages.

Response: Thank you for kind comment. We have revised and shorten the conclusions to make clearly (P12, L351-363).

Round 2

Reviewer 3 Report

The manuscript foods-956132 by Min-Pei Ling and co-authors was corrected but it still requires further improvements. Detailed comments are listed below.

  1. The English should be revised by professional services.
  2. Line 17: this sentence has no sense. What does “the risk of intake” mean?
  3. The literature still requires expansion because important studies related to phosphorus and its intake were omitted. These studies are e.g. Borgi, 2019; https://doi.org/10.2215/CJN.07230618; Hwang et al., 2015 13050/foodengprog.2015.19.2.161. In the revised version only 2 references were added and they apply to meat products only.
  4. The conclusion needs further improvements. Please add details (values) on the intake of phosphorus in Taiwanese population.
  5. Lines 76-79: please correct the sentence to avoid repetition.
  6. Figure 1: please add numbers of bought samples to all levels in the Figure 1.
  7. Paragraphs 2.2 and 2.3 must be shorten.
  8. Figure 4: add an information that the numbers form Figure 4 correspond to numbers in the Supplementary Material.

Author Response

This manuscript is a resubmission of an earlier submission. The following is a list of the peer review reports and author responses from that submission.

Round 1

Reviewer 1 Report

Brief summary

I would like to congratulate authors for this researching work. It is a well-designed study, and, until now,  there is few papers like this. However, in my opinion there are some considerations that must be explained.

Broad comments

INTRODUCTION. In my opinion the last paragraph must be expressed differently. The objectives of the study must be stated in a concise and clear way. How these objectives are to be achieved corresponds to the material and methods section. I think that it is confusing and difficult for the reader to understand.

MATERIAL AND METHODS. Inclusion and exclusion criteria of survey participants have to be expressed. Is someone with chronic renal disease or medical diet?

In 2.2 title, different scenario paragraph must to be explained clearly.

How calcium intake was calculated have to be discussed.

RESULTS AND DISCUSSION. Data about different subgroups needs to be improved. I think there is a lot of difference between 0-6 months, 6-12 months and between 1-3 years and 4-9 years, because food style is completely different, especially in toddlers. It is necessary add some data and tables to make it easier to understand.

CONCLUSIONS. It is well expressed and responding study’s objectives.

Specific comments

Line 5-9: There are no affiliations for Yi-Ting Kao. Moreover, there is nobody with the affiliation number 2.

Line 56-75: must be expressed in other way.

Line 76-83: there is an application form mistake. It corresponds to instructions for authors.

Line 88. Inclusion and exclusion criteria.

Line 109-134. Must be clarified.

Line 155. First time of LOD in text. Abbreviation explanation must be here instead of in line 156.

Line 197-199 and Line 216-217: Calcium and phosphorus intake for young people sounds excessive, according to EFSA Panel (EFSA Journal 2013;11(10):3408) and ietary Reference Intakes (DRIs): Recommended Dietary Allowances and Adequate Intakes, Elements (SOURCES: Dietary Reference Intakes for Calcium, Phosphorous, Magnesium, Vitamin D, and Fluoride (1997); Dietary Reference Intakes for Thiamin, Riboflavin, Niacin, Vitamin B6, Folate, Vitamin B12, Pantothenic Acid, Biotin, and Choline (1998); Dietary Reference Intakes for Vitamin C, Vitamin E, Selenium, and Carotenoids (2000); Dietary Reference Intakes for Vitamin A, Vitamin K, Arsenic, Boron, Chromium, Copper, Iodine, Iron, Manganese, Molybdenum, Nickel, Silicon, Vanadium, and Zinc (2001); Dietary Reference Intakes for Water, Potassium, Sodium, Chloride, and Sulfate (2005); Dietary Reference Intakes for Calcium and Vitamin D (2011); and Dietary Reference Intakes for Sodium and Potassium (2019). https://www.ncbi.nlm.nih.gov/books/NBK545442/table/appJ_tab3/?report=objectonly.). It could be caused by putting different ages in the same group.

Figure 2: There is no differences between adequate intakes for all subgroups?

Figure 4: 7-10 years group is no represented.

Table 1: AI, DRI are completely different between 0-6m , 6-12m and 12-24m. I think that it must be added more subgroups.  %MTD is not the same for a toddler 3 months old and for a preschool child 22 months old.

275-281: I think it is no comparable toddlers with a phosphorus intake of 6,6g. According to reference 21: “The results of the 2011 survey on the position and nutritional status of infants and young children in Taiwan showed that the average daily intake of 1-3 year old children is 1355 calories, protein 47.4 grams (14% of total calories), fat 35.2 grams (23.4%), carbohydrate 216.8 grams (64%)…” it is different that mean intake of toddlers 1-24m and I think that must be done age subgroups. Authors hypotheses that in toddlers there are no problems with the excessive intake of phosphorus but normal kcal intake for toddlers (first year of life) varies for 95 and 70 kcal/kg/day.  On the other hand, for healthy adults, the Institute of Medicine recommends a dietary intake of 700 mg/day of phosphorus, with an upper limit of 4 g/day (source: Institute of Medicine: Dietary reference intakes: Elements.).

310: calcium:phosphorus ratio is only presented in the elderly subgroup. It might be of interest to add a table with all the ratios.

360-362: instructions for authors. Mistake.

376-377. Authors write:  “For the intake of calcium, only the toddler group aged 1 to 2 years and children aged 3 to 9 years meet the AI”, but this is no represented in the figure 2 for children 3-9.

Reviewer 2 Report

Ling et al have performed a study called 'Total Dietary Study and Risk Assessment for  Estimating Dietary Intake of Phosphate in the Taiwanese Population'.

It is an interesting study, however I have the following comments on the manuscript.

First of all: The title of the manuscript should be changed and made clearer.

Line 72. The statement on the declining bone density with age should have a reference.

Lines 76-83. The paragraph is not part of the study and should be removed.

Lines 56-75. The aims of the study is written in a very long paragraph that is a bit difficult to follow. This paragraph should be shortened and aims made more clear for the readers.

Line 91. The 16 food categories that the data was clustered in should be mentioned already here.

Lines 179-80. The sentence starting with " The use amount of food additive...." seems to be unfinished. Please check.

Lines 190-191. A reference for the Dietary reference intakes of Taiwanese should be included.

Lines 212-213. The sentence regarding widespread use of phosphates should be elaborated on. What widespread use?

Line 234. The word diary is incorrect and should read Dairy, please correct. The same spelling error is found in Figure 3.

Lines 286-293. I doubt that the literature on the dietary P/Ca ratio and bone health is as clear cut as depicted in the manuscript. Please elaborate on the association between P/Ca and bone health.

Lines 294-303. The % are shown as for example 69.29 %. Is the data that certain. Please consider to round off.  The use of these exact numbers should be checked throughout the manuscript.

The figure legend for Figure 5 does not make sense. Please revise. The abbreviation RDA is used in the legend should be introduced before it is used. The figure does not depict histograms but box plots.

Lines 308-322. The paragraph needs some revision. Is there a recommended P/Ca-ratio? Please state this and give a proper reference and discuss around this. 

Line 317. RDA is not the abbreviation for recommended dietary intake, please correct.

Lines 320-322 lacks references.

Lines 336-338. How does mortality come into the picture here?

Lines 360-362 should be taken out of the manuscript.

Reviewer 3 Report

title of the paper is interesting, more time needs to be spent on the presentation in the paper from the whole paper,  from intro to Dis.